# A paternal bias in germline mutation is widespread in amniotes and can arise independently of cell division numbers

Marc de Manuel[1]*[†], Felix L Wu[1]*[†], Molly Przeworski[2]*

[1]Department of Biological Sciences, Columbia University, New York, United States; [2]Department of Systems Biology, Columbia University, New York, United States

*For correspondence:
md3914@columbia.edu (MdM);
flw2113@cumc.columbia.edu
(FLW);
mp3284@columbia.edu (MP)

[†]These authors contributed
equally to this work

Competing interest: See page
19

Reviewing Editor: Yukiko M
Yamashita, Whitehead Institute/
MIT, United States

**Abstract** In humans and other mammals, germline mutations are more likely to arise in fathers than in mothers. Although this sex bias has long been attributed to DNA replication errors in spermatogenesis, recent evidence from humans points to the importance of mutagenic processes that do not depend on cell division, calling into question our understanding of this basic phenomenon. Here, we infer the ratio of paternal-to-maternal mutations, $\alpha$, in 42 species of amniotes, from putatively neutral substitution rates of sex chromosomes and autosomes. Despite marked differences in gametogenesis, physiologies and environments across species, fathers consistently contribute more mutations than mothers in all the species examined, including mammals, birds, and reptiles. In mammals, $\alpha$ is as high as 4 and correlates with generation times; in birds and snakes, $\alpha$ appears more stable around 2. These observations are consistent with a simple model, in which mutations accrue at equal rates in both sexes during early development and at a higher rate in the male germline after sexual differentiation, with a conserved paternal-to-maternal ratio across species. Thus, $\alpha$ may reflect the relative contributions of two or more developmental phases to total germline mutations, and is expected to depend on generation time even if mutations do not track cell divisions.

## Editor's evaluation

This paper challenges a fundamental view concerning why males of most animals have a higher germline mutation rate than females. Evidence is provided to show that it is not simply the fact that males have more cell divisions in the germline, but instead, most of the mutations arise from a different balance of DNA damage vs. DNA repair. The case is supported by data from multiple species, from de novo mutation rate estimates from pedigrees, and from fits to a simple heuristic model. This work will be of interest to the broad field of DNA mutations and DNA repair, as well as evolutionary and phylogenomics researchers.

## Introduction

Humans tend to inherit more de novo mutations (DNMs) from their fathers than from their mothers. This phenomenon was first noted over 70 years ago, when JBS Haldane relied on the population frequency of hemophilia in order to infer that the DNM rate at the disease locus is substantially higher in fathers (*Haldane, 1946*). Work since then, particularly in molecular evolution, has confirmed a 'male bias' in mutation (henceforth paternal bias) (*Makova and Li, 2002*; *Wolfe and Li, 2003*; *Li et al., 1996*; *Presgraves and Yi, 2009*; *Nachman and Crowell, 2000*; *Huang et al., 1997*; *Shimmin et al., 1993b*; *Chang et al., 1994*), with estimates from human pedigrees indicating that, genome-wide, DNMs occur roughly four times more often on the paternal genome than on the maternal one (*Kong et al., 2012*; *Francioli et al., 2015*).

The textbook explanation for the paternal mutation bias is that it arises as a consequence of the vastly different numbers of cell divisions – and hence DNA replication cycles – necessary to produce sperm compared to oocytes (*Crow, 2000*; *Drost and Lee, 1995*; *Penrose, 1955*; *Strachan and Read, 2018*). In humans as in other mammals, oocytes are arrested in meiotic prophase I at birth, with no subsequent DNA replication in the mother's life, whereas spermatogonia start dividing shortly before puberty and divide continuously throughout the reproductive life of the father (*Drost and Lee, 1995*; *Guo et al., 2020*). The observation that the number of DNMs increases with paternal age has been widely interpreted in this light, as evidence for DNA replication errors being the predominant source of germline mutation (*Kong et al., 2012*; *Francioli et al., 2015*; *Goldmann et al., 2019*; *Jónsson et al., 2017*).

A number of recent findings have called this view into question, however. First, analyses of large numbers of human pedigrees revealed an effect of maternal age on the number of maternal DNMs (*Wong et al., 2016*; *Goldmann et al., 2016*), with an additional ~0.4 mutations accrued per year. Given the lack of mitotic cell division in oocytes after birth, this observation indicates that by typical reproductive ages, at least half of maternal DNMs arise from DNA damage (*Jónsson et al., 2017*). Second, despite highly variable rates of germ cell division over human ontogenesis, germline mutations appear to accumulate with absolute time in both sexes, resulting in a ratio of paternal-to-maternal germline mutation, $\alpha$, of around 3.5 at puberty and very little increase with parental ages (*Gao et al., 2019*). Third, studies in a dozen other mammals suggest that $\alpha$ ranges from 2 to 4 whether the species reproduces months, years, or decades after birth (*Wu et al., 2020*; *Wilson Sayres et al., 2011*; *Wang et al., 2022a*), when estimates of germ cell division numbers at time of reproduction would predict a much wider range in $\alpha$ (*Drost and Lee, 1995*; *Lindsay et al., 2019*; *Harland et al., 2017*; *Wu et al., 2020*).

Explaining the observations in humans under a model in which most mutations are due to replication errors, and thus track cell divisions, would call for an exquisite balance of cell division and mutation rates across developmental stages in both sexes (*Gao et al., 2016*). In males, the constant accumulation of mutations with absolute time would require varying rates of germ cell divisions over ontogenesis to be precisely countered by reciprocal differences in the per cell division mutation rates. In females, it would necessitate that the mutation rate per unit of time be identical whether mutations arise from replication errors or damage. In turn, the similarity of $\alpha$ across mammals that differ drastically in their reproductive ages would entail two distinct sources of mutation – replication error in males and damage in females – covarying in tight concert with generation times.

A more parsimonious alternative is that most germline mutations arise from the interplay between damage and repair rather than from replication errors (*Seplyarskiy et al., 2021*), and that the balance results in more mutations on the paternal than the maternal genome (*Gao et al., 2016*). Assuming repair is inefficient relative to the length of the cell cycle or, perhaps more plausibly, that repair is efficient but inaccurate (*Vilenchik and Knudson, 2003*; *Abascal et al., 2021*), mutations that arise from damage will not track cell divisions (*Gao et al., 2016*). Damage-induced mutations must underlie the observed maternal age effect on DNMs in humans; they could also account for the accumulation of germline mutations in proportion to absolute time in males, assuming fixed rates of damage and repair machinery errors in germ cells.

Multiple lines of evidence have emerged in support of damage-induced mutations being predominant in the human germline. Analyses of the mutation spectrum in humans indicate that 75% of DNMs and 80% of mutations in adult seminiferous tubules are due to mutation 'signatures' SBS5/40 (*Rahbari et al., 2016*; *Moore et al., 2021*), which are clock-like, uncorrelated with cell division rates in the soma (*Alexandrov et al., 2015*; *Alexandrov et al., 2020*), and also predominant in post-mitotic cell types such as neurons (*Lodato et al., 2018*; *Abascal et al., 2021*). In addition, most substitutions in post-pubertal germ cell tumors are attributed to SBS5/40, in both females and males (*Oliver et al., 2022*). More generally, cell division rates do not appear to be a major determinant of mutation rates across somatic tissues (*Blokzijl et al., 2016*): notably, post-mitotic neurons accumulate mutations at a similar rate as mitotic somatic cell types that are the product of ongoing cell divisions (*Abascal et al., 2021*). A decoupling between cell division numbers and mutation burden has also been described in colonic crypts across mammals (*Cagan et al., 2022*), and in yeast, up to 90% of mutations have been estimated to be non-replicative in origin (*Zhou et al., 2021*). Altogether, these results suggest an important role, for both germline and soma, of

mutagenic processes that accumulate with absolute time, as expected from damage-induced mutations (*Gao et al., 2016*).

In undermining the prevailing understanding of the paternal bias in human germline mutations, these observations revive the question of how the bias arises, as well as of the influences of life history traits and exogenous or endogenous environments. To investigate them, we took a broad taxonomic view, characterizing the paternal mutation bias across amniotes, including mammals but also birds and snakes, which differ in potentially salient dimensions. As two examples, in birds as in mammals, oogenesis is arrested by birth in females, while spermatogenesis is ongoing throughout male reproductive life (*Guraya, 1989*; *Deviche et al., 2011*), but birds have internal testes whereas mammals usually have external testes. In addition, mammals and birds are endotherms, in contrast to ectothermic reptiles such as snakes. More generally, the taxa considered vary widely in their life histories, physiologies, and natural habitats.

## Results

### Estimating sex differences in germline mutation rates across amniotes

To estimate $\alpha$ in each lineage, we based ourselves on the evolutionary rates at putatively neutrally evolving sites of sex chromosomes compared to the autosomes (*Miyata et al., 1987*). The more direct approach of detecting DNMs in pedigrees requires them to be available for each species, and in large numbers for the estimates to be precise. In contrast, the evolutionary method is in principle applicable to any set of species with high-quality genome assemblies and a stable sex karyotype. It takes advantage of the fact that at the population level, sex chromosomes spend different numbers of generations in each sex (e.g., the X chromosome spends twice as many generations in females as in males), whereas autosomes spend an equal number in both (*Figure 1A*). Thus, all else being equal, if there is a paternal mutation bias, an autosome with greater exposure to the more mutagenic male germline will accumulate more neutral substitutions than the X over evolutionary timescales (*Figure 1A*); the inverse will be true for the autosomes compared to the Z chromosome (*Miyata et al., 1987*).

Such evolutionary approaches have been widely applied, but until recently they were limited in the number of loci or species (e.g., *Shimmin et al., 1993b*; *Huang et al., 1997*; *Pecon Slattery and O'Brien, 1998*; *Ellegren and Fridolfsson, 1997*; *Carmichael et al., 2000*; *Nachman and Crowell, 2000*) and did not take into account the influence of sex differences in generation times on the estimation of $\alpha$ (*Wilson Sayres et al., 2011*). An additional complication to consider is that X (Z) and autosomes differ not only in their exposures to male and female germlines but in a number of technical and biological features (notably, GC content) that may need to be controlled for (*Shimmin et al., 1993a*; *Pink and Hurst, 2010*; *Agarwal and Przeworski, 2019*). Moreover, analyses involving closely related species can be confounded by the effects of ancestral polymorphism: for example, lower ancestral diversity in the X chromosome relative to the autosomes reduces the X-to-autosome divergence ratio, leading to overestimation of $\alpha$ (*Presgraves and Yi, 2009*; *Figure 1B*). In birds, unresolved branches within the phylogeny present an additional difficulty in estimating substitution rates (*Jarvis et al., 2014*; *Reddy et al., 2017*).

Here, we designed a pipeline for estimating the paternal mutation bias systematically across a wide range of species, mindful of these issues. To these ends, we employed existing whole genome alignments (*Zoonomia Consortium, 2020*; *Feng et al., 2020*) or produced our own (for snakes, see Sequence alignments in Materials and methods), focusing on assemblies with high quality and contiguity and, where possible, those based on a homogametic individual. To handle the confounding effects of ancestral polymorphism on divergence, we thinned species in the phylogeny to ensure a minimum level of divergence between them, relative to polymorphism levels (see Species selection criteria in Materials and methods). This stringent filtering procedure resulted in three whole genome alignments including 20 mammals, 17 birds, and 5 snake species, respectively (*Supplementary file 2*).

In order to estimate neutral substitution rates from the alignments and compare X (Z) and autosomes while minimizing confounding factors, we focused on non-repetitive, non-exonic regions that were orthologous across all species in an alignment and did not overlap with pseudo-autosomal regions (PARs) with orthologs on the Y (W) chromosome (see Selecting non-repetitive and putatively neutral sequences in Materials and methods; see *Figure 2—figure supplement 1F* for a more stringent masking of all conserved regions). To account for differences between X (Z) and autosomes in

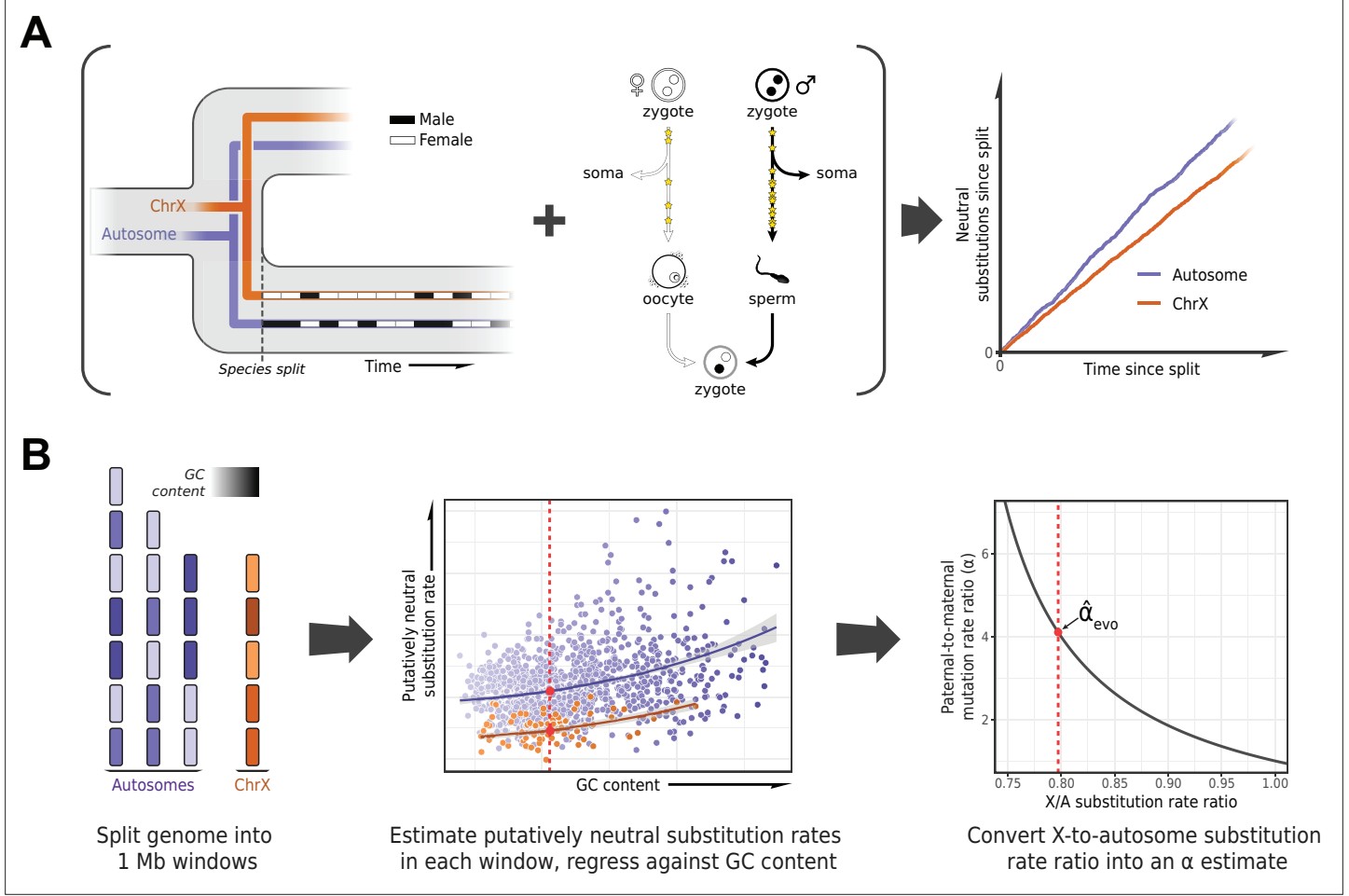

**Figure 1.** Estimating the paternal bias in mutation from neutral substitution rates of sex chromosomes and autosomes. (**A**) On average, the lineage of an X chromosome spends fewer generations in males than females. Given a higher mutation rate in males than in females and all else being equal, this leads to lower rates of neutral substitutions on the X chromosome compared to autosomes (*Miyata et al., 1987*). (**B**) Procedure for estimating the ratio of paternal-to-maternal mutation rates, α, from substitution rates in sex chromosomes and autosomes. The autosomes and the X chromosome are partitioned into 1 Mb windows, depicted in purple and orange, respectively. Each window is filtered to focus on putatively neutrally evolving sequences (see Selecting non-repetitive and putatively neutral sequences in Materials and methods), and its GC content is calculated (represented by shading). The putatively neutral substitution rates per window are then regressed against the GC content (center panel, see Estimating α from X-to-autosome substitution rate ratios in Materials and methods). Substitution rate estimates for the X chromosome and autosomes are obtained from the regression fit (red points). Finally, the ratio of the point estimates is converted to an estimate of α (right panel). An analogous procedure applies to comparisons of the Z chromosome and autosomes in a ZW sex determination system.

The online version of this article includes the following figure supplement(s) for figure 1:

**Figure supplement 1.** Identification of pseudo-autosomal regions in *Thamnophis*.

features other than their exposure to each sex, we regressed putatively neutral substitution rates in the 1 Mb genomic windows against GC content and GC content squared (*Figure 1B*). We took this approach because GC content is readily obtained from any genome sequence and is highly correlated with known modifiers of the mutation rate such as replication timing and the fraction of CpG dinucleotides (*Koren et al., 2012*; *Agarwal and Przeworski, 2019*). In principle, X chromosome inactivation could also influence relative substitution rates on X versus autosomes, but in the germline, it is short-lived: limited in mice and humans to early embryogenesis in females and brief meiotic and post-meiotic periods in males (*Chuva de Sousa Lopes et al., 2008*; *Guo et al., 2015*). We obtained substitution rate estimates for the X (Z) chromosome and autosomes from the regression fit. Finally, we inferred α for the terminal branches leading to the 42 amniote species from the ratio of the substitution rate estimates for the X (Z) versus the autosomes (*Figure 2*), taking into account sampling error

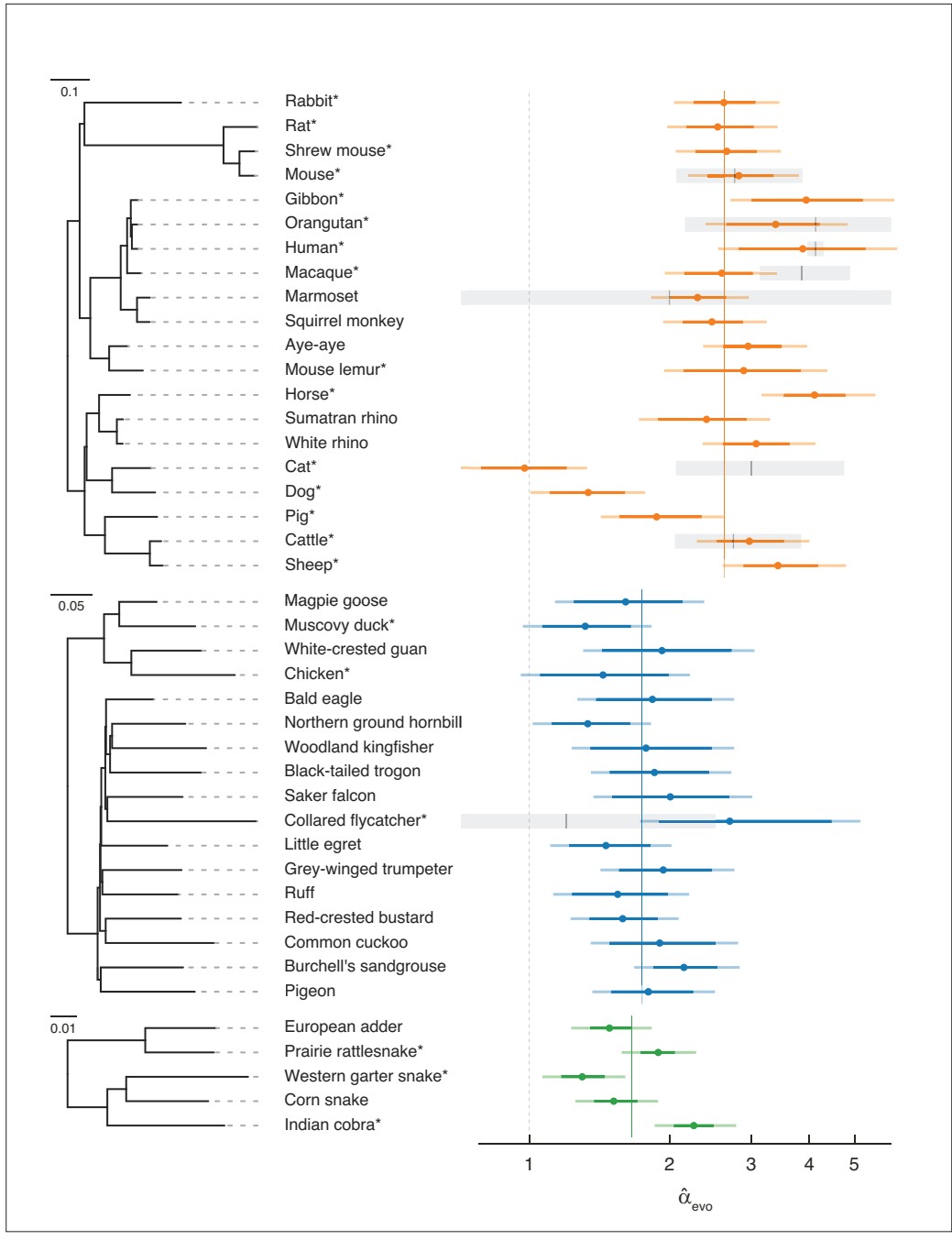

**Figure 2.** Estimates of the paternal bias in mutation across 42 amniote lineages. Colored points denote estimates of $\alpha$ from X (Z)-to-autosome substitution rate ratios ($\hat{\alpha}_{\text{evo}}$) in mammals (top, orange), birds (middle, blue), and snakes (bottom, green). Vertical colored lines denote the mean $\hat{\alpha}_{\text{evo}}$ for each group, while the vertical gray dashed line denotes $\alpha = 1$ (i.e., no sex bias in mutation). Species in each group are plotted by their phylogenetic relationships and branch lengths are scaled by the neutral substitution rate estimated from autosomes (see Estimating putatively neutral substitution rates in Materials and methods). Note that branch lengths are comparable within the phylogeny of each taxon but not across taxa, as the scaling differs (see the legend for each group). In mammals, $\hat{\alpha}_{\text{evo}}$ was estimated from neutral substitutions along the lineage from the tip to the most recent common ancestor indicated in the phylogeny. In birds, where phylogenetic relationships are more tenuous, we divided species into six subgroups (***Supplementary file 5***) to avoid highly uncertain ancestral nodes in Neoaves; thus, some $\hat{\alpha}_{\text{evo}}$ estimates in Neoaves average over deeper splits than suggested by the full phylogeny, which we plot for clarity. Asterisks indicate species with chromosome-level assemblies. Darker colored horizontal lines behind the points represent 95% CIs, which were computed by bootstrap resampling of the 1 Mb genomic

*Figure 2 continued on next page*

*Figure 2 continued*

windows across 500 replicates; the central 95% interval across bootstrap replicates is shown. Lighter colored horizontal lines include uncertainty in the ratio of paternal-to-maternal generation times, allowing the ratio to range between 0.9 and 1.1 (*Amster and Sella, 2016*). Short vertical red lines denote point estimates of $\hat{\alpha}_{dnm}$ from published pedigree mutation studies of de novo mutations, and the surrounding horizontal gray boxes represent the 95% binomial CI for those estimates.

The online version of this article includes the following figure supplement(s) for figure 2:

**Figure supplement 1.** $\hat{\alpha}_{evo}$ for each species, obtained under variants of the pipeline presented in the main text.

**Figure supplement 2.** Expected equilibrium GC content (GC*) in the mammalian X chromosomes.

**Figure supplement 3.** Estimation of $\hat{\alpha}_{evo}$ for mutation types affected or unaffected by GC-biased gene conversion (gBGC).

---

as well as uncertainty in the ratio of paternal-to-maternal generation times (*Amster and Sella, 2016*) (see Estimating $\alpha$ from X-to-autosome substitution rate ratios in Materials and methods).

Overall, our evolutionary-based estimates, $\hat{\alpha}_{evo}$, are consistent with estimates from pedigree sequencing studies, $\hat{\alpha}_{dnm}$ (*Figure 2*). Notably, and reassuringly, the point estimates for species with the largest amount of available DNM data (e.g., humans, mice, and cattle) are in very close agreement. Even in the absence of estimation error, this concordance is not necessarily expected, as $\hat{\alpha}_{evo}$ is an average over many thousands of generations of evolution, whereas estimates from DNMs are based on small numbers of families at present. In principle, differences between the estimates could therefore arise if $\alpha$ evolves rapidly (as may have happened in the lineage leading to macaque), or if the ages of the parents in the pedigree are quite unrepresentative of average paternal-to-maternal generation times in evolution (*Figure 2*; *Amster and Sella, 2016*). The general concordance between $\hat{\alpha}_{evo}$ and $\hat{\alpha}_{dnm}$ therefore suggests that the evolutionary approach is providing reliable estimates and the paternal bias in mutation is not rapidly evolving.

Nonetheless, it is unlikely that our regression model perfectly accounts for all the genomic features that differ between sex chromosomes and autosomes other than exposure to sex. Remaining disagreement between $\hat{\alpha}_{evo}$ and $\hat{\alpha}_{dnm}$ could therefore also arise from mutation rate modifiers that differentially affect sex chromosomes and autosomes. For example, in cats, the low $\hat{\alpha}_{evo}$ compared to $\hat{\alpha}_{dnm}$ (*Wang et al., 2022b*) could be due to unusual features of the X chromosome: the feline X chromosome is known to harbor a large recombination coldspot spanning over 50 Mb (*Li et al., 2016*), visible in its effects on GC substitution rates (*Figure 2—figure supplement 2*; *Meunier and Duret, 2004*), and these features may have influenced the rate of substitution of the X chromosome relative to the autosomes.

## A paternal bias in mutation is widespread in amniotes

A paternal bias in mutation is seen across amniotes, with a range of 1–4 in the species considered (*Figure 2*). The $\hat{\alpha}_{evo}$ estimates remain similar if we exclude hypermutable CpG sites (*Figure 2—figure supplement 1B*), or focus only on mutation types that are not subject to the effects of GC-biased gene conversion (gBGC) (*Figure 2—figure supplement 1F* and *Figure 2—figure supplement 3*). Although the absolute magnitude of $\hat{\alpha}_{evo}$ exhibits some sensitivity to different choices of conservation filters (e.g., excluding all conserved regions, not just exons) and different substitution types, $\hat{\alpha}_{evo}$ are robustly above 1 and their ranking across species remains similar across different filtering criteria (see *Figure 2—figure supplement 1* for details). The robustness of $\hat{\alpha}_{evo}$ across conditions and filters confirms that, while our pipeline may not account for all the differences between autosomes and X (Z) chromosomes unrelated to sex differences in mutation, the qualitative patterns are reliable. These results therefore establish that the paternal bias in mutation is not a feature of long-lived humans or of mammals, but is instead ubiquitous across species that vary markedly in their gametogenesis, physiology, and life history.

The effects of gBGC track recombination rates and result in greater selection for GC in regions of higher recombination. Therefore, if $\alpha$ is similar for different types of DNMs, as has been found in humans (*Jónsson et al., 2017*; *Gao et al., 2019*), the greater population recombination rate of autosomes relative to the sex chromosomes should lead the the X-to-autosome substitution rate ratio of gBGC-favored mutation types (T>C and T>G) to be somewhat lower than that of mutation types

unaffected by gBGC (C>G and T>A). Consistent with expectation, $\hat{\alpha}_{\text{evo}}$ estimates in mammals using only gBGC-favored mutation types are inflated relative to estimates from mutation types unaffected by gBGC (*Figure 2—figure supplement 3*). Also as expected, bird and snake species with ZW sex determination exhibit the opposite pattern (i.e., a deflated ratio of Z-to-autosome substitution rate leads to a decreased estimate of $\hat{\alpha}_{\text{evo}}$; *Figure 2—figure supplement 3*). The behavior of the different mutation types therefore provides a further sanity check on our estimates. While the estimation of $\hat{\alpha}_{\text{evo}}$ could be further partitioned into single mutation classes, such estimates are noisier and – given the lack of ground truth – harder to interpret; we therefore focused on $\alpha$ for all substitution types combined.

Within mammals, the mean value of $\hat{\alpha}_{\text{evo}}$ is 2.7, with a range 1.0–4.1 and a coefficient of variation of 0.29. In birds, $\hat{\alpha}_{\text{evo}}$ is lower on average but also seemingly more stable, ranging from 1.5 to 2.7 (mean

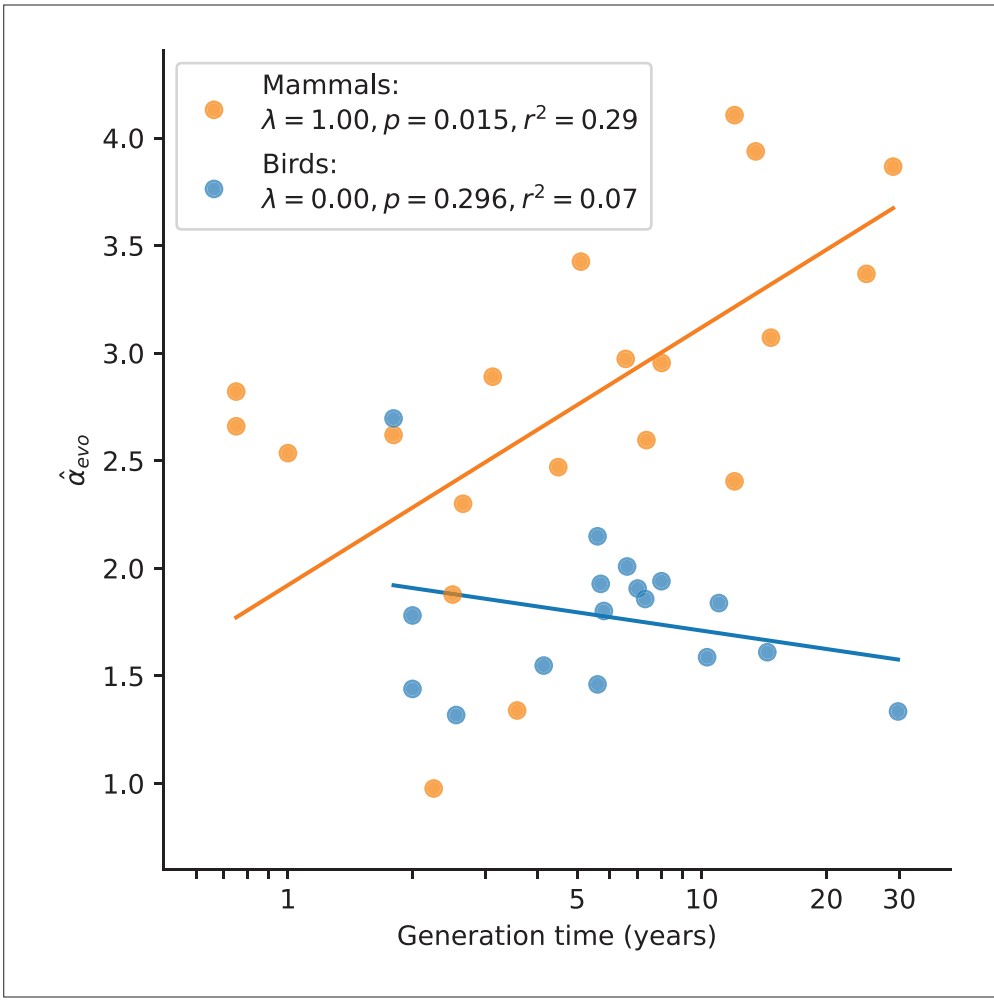

**Figure 3.** Relationship between $\hat{\alpha}_{\text{evo}}$ and generation time estimates in mammals and birds. Estimates of $\alpha$ from X (Z)-to-autosome comparisons are plotted against generation times from the literature (see *Supplementary file 2*), on a log scale. Lines denote the phylogenetic generalized least squares regression fits in mammals (orange) and birds (blue). $\lambda$ refers to Pagel's $\lambda$ (*Pagel, 1999*), a measure of the strength of phylogenetic signal, which was inferred via maximum likelihood (see Testing relationships between $\alpha$ and life history traits in Materials and methods). Fixing $\lambda$ to 1 in birds, as estimated for mammals, did not meaningfully improve the fit (p-value =0.282, $r^2 = 0.08$).

The online version of this article includes the following figure supplement(s) for figure 3:

**Figure supplement 1.** Relationship between mammalian $\hat{\alpha}_{\text{evo}}$ and various life history traits.

**Figure supplement 2.** Principal component (PC) analysis of four life history traits.

= 1.8, coefficient of variation = 0.19). In the handful of snake species sampled, the mean is similar to that of birds and $\hat{\alpha}_{\text{evo}}$ ranges from 1.3 to 2.2 (mean = 1.7, coefficient of variation = 0.23), in agreement with a previous evolutionary estimate for rattlesnake ($\alpha$=2.0; *Schield et al., 2019*).

In mammals, variation in $\alpha$ has long been known to be associated with generation times, and has been consistently interpreted as resulting from greater numbers of replication errors in species with longer-lived fathers (e.g., *Wilson Sayres et al., 2011*; *Chang et al., 1994*; *Li et al., 1996*; *Li et al., 2002*). We confirmed the observation here: after accounting for the phylogenetic relationship between species, mammals reproducing at older ages show a stronger paternal bias in mutation (p-value = 0.01, $r^2$ = 29%; *Figure 3*). Statistically significant relationships also exist between $\hat{\alpha}_{\text{evo}}$ and other life history traits (*Figure 3—figure supplement 1*), but these traits are strongly correlated with one another (*Figure 3—figure supplement 2*) and generation time is the strongest single predictor (*Figure 3—figure supplement 1*; see Testing relationships between $\alpha$ and life history traits in Materials and methods). In contrast, a significant relationship between generation time and $\hat{\alpha}_{\text{evo}}$ is not seen in birds (p-value = 0.30, $r^2$ = 7%; *Figure 3*; *Wang et al., 2014*), despite similar numbers of species and a comparable range of generation times to mammals. Moreover, we could reject the null model of a slope in birds equal to or greater than that of mammals (p-value = $10^{-5}$). (Given the paucity of generation time and $\alpha$ estimates for snakes, we could not test the relationship in reptiles.) Given recent evidence that most mutations depend on absolute time and not cell division rates, the standard explanation for this generation time effect no longer holds. These observations therefore raise the question of how else the relationship between generation times and $\alpha$ in mammals can be explained.

## A cell-division-independent explanation for the correlation between $\alpha$ and generation time

In eutherian mammals, embryo development is likely independent of sex until primordial germ cell (PGC) specification and subsequent development of the gonads (*Lin and Capel, 2015*). As a result, mutations arising during early embryogenesis (*Early*) are expected to occur at a similar rate in males and females ($\alpha_{Early} = 1$), as has been inferred in the few pedigree studies in which DNMs during parental early embryogenesis are distinguished from mutations later in development, namely in humans (*Sasani et al., 2019*), cattle (*Harland et al., 2017*), and mice (*Lindsay et al., 2019*; *Figure 4A*). While sex differences in early development may exist (*Engel, 2018*), differences in male and female mutation rates at such an early stage are likely modest in mammals (*Spiller et al., 2017*; *Hancock et al., 2021*). At some point after sexual differentiation of the germline, however (in what we term the *Late* stage) mutation rates in the two sexes need no longer be the same: sources and rates of DNA damage could differ between germ cells, as could the efficiency and accuracy of repair. Indeed, human fathers that recently reached puberty contribute over three times more mutations than similarly aged mothers (*Gao et al., 2019*). Intriguingly, the magnitude of paternal bias for mutations that occurred long after sexual differentiation of the PGCs appears to be similar in mice, cattle, and humans, at approximately 4:1 (*Lindsay et al., 2019*; *Harland et al., 2017*; *Sasani et al., 2019*; *Figure 4A*).

In light of these observations, we considered a simple model in which $\alpha$ in mammals is the outcome of two developmental stages with distinct ratios of paternal-to-maternal mutations. In the *Early* stage until germline sex differentiation, we assumed a paternal-to-maternal mutation ratio of 1 and an expected number of mutations ($M_e$) on par with what is observed in humans (i.e., 5 mutations per haploid genome; *Sasani et al., 2019*; *Jonsson et al., 2021*; *Ju et al., 2017*; *Figure 4B*). In the *Late* developmental stage after germline sex differentiation, which varies in length among species, we assumed mutations arise at a constant rate per year, $\mu_s$ in sex $s$ ($s \in \{f, m\}$). If we assume the length of *Early* to be negligible relative to the generation time, $G_s$ in sex $s$, then the expectation of $\alpha$ can be written as:

$$\alpha = \frac{M_e + \mu_m G_m}{M_e + \mu_f G_f}.$$

(1)

If the ratio $\mu_m/\mu_f$ is 4 across species, as suggested by DNM data (*Lindsay et al., 2019*; *Harland et al., 2017*; *Sasani et al., 2019*; *Figure 4A* and *Supplementary file 3*), this model yields a relationship between $\alpha$ and generation time bounded below by 1 and with a plateau at 4, assuming the same generation times in the two sexes (*Figure 4C*); more generally, the height of the plateau depends on the ratio of paternal-to-maternal generation times (*Figure 4—figure supplement 1*). The rapidity

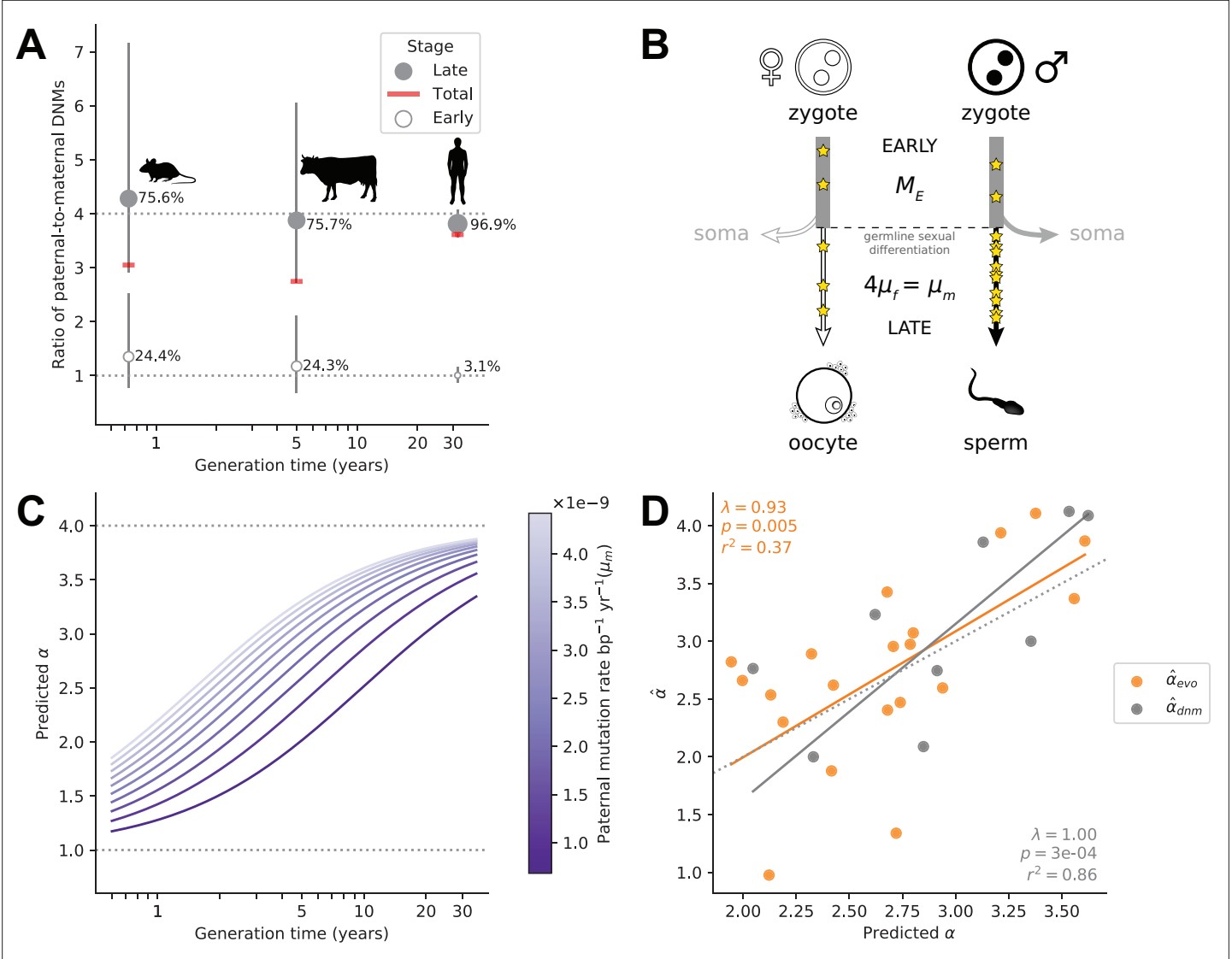

**Figure 4.** Variation in α among mammals may reflect varying exposures to different developmental stages. (**A**) Ratio of paternal-to-maternal de novo mutations (DNMs) occurring in early embryogenesis (*Early*, white points), after the sexual differentiation of the germline (*Late*, gray points) and in both of these stages combined (*Total*, red line), for the three mammalian species in which this classification is available (mouse *Lindsay et al., 2019*, cattle *Harland et al., 2017*, and human *Sasani et al., 2019*). For each species, the percentage of DNMs occurring at each stage are indicated and used to scale the size of points. Vertical lines show the 95% binomial CIs. Since the phasing rate is not equal across developmental stages, point estimates for α in *Total* were computed by extrapolating the proportion of paternally and maternally phased DNMs in each stage to all the DNMs in that stage (i.e., assuming full phasing) (see Estimating α from pedigree studies in vertebrates in Materials and methods). (**B**) Schematic representation of a model in which α is the outcome of mutation in two developmental stages (see Modeling the effects of germline developmental stages on α in Materials and methods). (**C**) Expected relationship between α and generation time under the model outlined in B, assuming generation times are the same in both sexes. The increase of α with generation time depends on the paternal mutation rate per year in *Late*, $\mu_m$, as illustrated by the purple gradient. (**D**) Fits of predicted α values to $\hat{\alpha}_{\text{evo}}$ (orange) and $\hat{\alpha}_{\text{dnm}}$ (gray). In each species, α is predicted with **Equation 1** assuming $M_e = 1.66 \times 10^{-9}$ and using $\mu_f$ and $\mu_m$, the latter estimated from autosomal branch-specific substitution rates per year ($\hat{\alpha}_{\text{evo}}$) or as estimated from pedigree sequencing data ($\hat{\alpha}_{\text{dnm}}$) (see Modeling the effects of germline developmental stages on α in Materials and methods). The orange and gray lines denote the regression fit using phylogenetic generalized least squares (PGLS). PGLS statistics are shown for the two models (see *Figure 3* legend for details).

The online version of this article includes the following figure supplement(s) for figure 4:

**Figure supplement 1.** The maximal value of α depends on the ratio of paternal-to-maternal generation times.

**Figure supplement 2.** Ratio of crypt-to-sperm mutation rate per unit of time in four mammals.

with which $\alpha$ reaches this asymptote is determined by the magnitude of $\mu_m$ (and $\mu_f$) in the *Late* stage (*Figure 4C*). Most pertinent, a positive relationship between $\alpha$ and the sex-averaged generation time is expected as long as $\mu_m G_m > \mu_f G_f$.

Using this model, we then predicted $\alpha$ for the terminal branches in the mammalian tree. To estimate the number of mutations occurring in *Late* for each branch, we used the evolutionary rates in *Figure 2A*. Specifically, we calculated a sex-averaged substitution rate per generation by multiplying the autosomal yearly substitution rate in each branch ($\mu_y$) by a generation time estimate for its tip (*Supplementary file 2*). Given a fixed ratio of paternal-to-maternal mutation rates of 4 in the *Late* stage, the mutation rate for each sex can be calculated for any given ratio of paternal-to-maternal generation times:

$$\mu_f = \frac{\mu_y \left( G_f + G_m \right) - 2M_e}{G_f + 4G_m}. \tag{2}$$

From the parental mutation rates and assuming a fixed $M_e$, we obtained an estimate of $\alpha$ that we can use to predict $\hat{\alpha}_{\text{evo}}$ using *Equation 1* (see Modeling the effects of germline developmental stages on $\alpha$ in Materials and methods). This model explains a significant proportion of the variance in $\hat{\alpha}_{\text{evo}}$ in mammals ($r^2 = 37\%$; p-value = 0.005; *Figure 4D*). After taking into account sampling error in our $\hat{\alpha}_{\text{evo}}$ estimates (see Modeling the effects of germline developmental stages on $\alpha$ in Materials and methods), it explains 42% of the variance in $\alpha$ across species. Moreover, the fit of the model remains good regardless of the precise number of *Early* mutations assumed (see Modeling the effects of germline developmental stages on $\alpha$ in Materials and methods). The two clear outliers are carnivores, for which $\hat{\alpha}_{\text{evo}}$ may be an underestimate, given the higher estimate from DNMs in cats (*Figure 2*).

These predictions rely on evolutionary estimates that are uncertain, due for instance to inaccuracies in split time estimates and the use of contemporary generation times as proxies for past ones. If we instead predict $\alpha$ using parameters derived from pedigree data in the nine mammalian species for which at least 30 DNMs have been phased and more than one trio has been studied (Modeling the effects of germline developmental stages on $\alpha$ in Materials and methods), the model explains 86% of the variance in $\hat{\alpha}_{\text{dnm}}$ (p-value = $3 \times 10^{-4}$; *Figure 4D*). We caution that this assessment is based on few phylogenetically independent contrasts, however, and so while the fit of the model again appears quite good, the variance explained may be deceivingly high.

In any case, this phenomenological model clarifies that the increased $\alpha$ seen in long-lived mammals may simply reflect a reduction in the fraction of early embryonic mutations relative to total number of mutations per generation – consistent with the higher proportion of *Early* mutations in mice and cattle compared to humans (*Figure 4A*). This model can also explain the only modest increase in $\alpha$ with parental ages observed in humans (*Gao et al., 2019*).

Given this explanation for the effect of generation times on $\alpha$ in mammals, why is a relationship not seen in birds (*Figure 2*)? One interpretation is simply a lack of statistical power: since the ratio of paternal-to-maternal age effects in the *Late* stage is lower in birds than in mammals (around 2 instead of 4), under our model, bird generation times would influence $\alpha$ within a narrower range (i.e., between 1 and 2). Alternatively, the lack of a relationship between $\alpha$ and generation times in birds could reflect their distinct germ cell development: Unlike mammals, avian sexual phenotype is directly determined by the sex chromosome content of individual cells (*Zhao et al., 2010*; *Ioannidis et al., 2021*) and PGCs are determined by inheritance of maternally derived gene products (*Extavour and Akam, 2003*). Given these features, it seems plausible that sex differences in mutation rates appear earlier in ontogenesis in birds than in mammals, consistent with reported sex differences in the cellular phenotypes of PGCs prior to gonad development (*Soler et al., 2021*). If indeed the mutation rate in the two bird sexes differs from very early on in development (i.e., if term $M_e \approx 0$ in *Equation 1*), then assuming a fixed ratio of paternal-to-maternal generation times, our model predicts the sex-averaged age of reproduction will have little to no influence on $\alpha$.

## Discussion

Analyzing diverse species with the same pipeline, we found that, far from being a feature of species with long-lived males, a paternal bias in germline mutation is ubiquitous across amniotes that differ markedly in their life history, physiology, and gametogenesis. Moreover, by considering the different

development stages over which germline mutations arise, we provide a new and simple explanation for variation in the degree of sex bias across mammals that does not require dependence on the number of cell divisions.

These findings do not explain *why* male germ cells accumulate more mutations than female ones, however. Given that the paternal bias varies little across species exposed to disparate physical environments, and presumably distinct exogenous mutagens, the proximate causes of the paternal bias are likely sex differences in endogenous sources of DNA damage or in repair mechanisms. For instance, the effects of reactive oxygen species, a major source of DNA damage, may be greater in male germ cells than in oocytes (*Smith et al., 2013*; *Rodríguez-Nuevo et al., 2022*). In turn, the evolutionary cause of the paternal bias could be related to the different evolutionary pressures acting on each sex of anisogamous species, for example due to greater competition among sperm than among oocytes.

Another question raised by our findings is why, after sexual differentiation of the germline, mutation appears to be more paternally biased in mammals (~4:1) than in birds and snakes (~2:1). In that regard, it will be of interest to collect pedigree data from these taxa, with which to compare mutation signatures to those typically seen in mammals.

Beyond these questions, our findings suggest a change of focus, reframing sex differences in germline mutation rate as part of a broader puzzle: why certain cell types accrue more mutations than others. In that regard, it is intriguing that the relative mutation rates of different cell types seem similar across mammals. The balance of damage and repair results in an approximately fourfold higher mutation rate in spermatogonia compared to oocytes across mammalian species (*Figure 4A*). Similarly, comparing yearly mutation rates in colonic crypts (*Cagan et al., 2022*) to estimates for spermatogonia, the ratio of crypt-to-sperm mutation rates appears relatively stable across four mammalian species (*Figure 4— figure supplement 2*). This observation suggests that, beyond spermatogonia and oocytes, the relative mutagenicity of different cell types may be conserved across mammals. Mutation rates in different cell types could be coupled over time either because of natural selection to maintain specific rates in each cell type or because changes to the repair machinery in some cell types (potentially, germ cells) have pleiotropic consequences on mutation rates in others. Regardless, our observations point to a role of natural selection in maintaining the relative rates at which mutations accumulate in different cell types over long evolutionary timescales.

## Materials and methods
### Sequence alignments

In mammals, we obtained sequence alignments from the 241-way multi-alignment generated by the Zoonomia Project (https://zoonomiaproject.org/) (*Zoonomia Consortium, 2020*). To assess the effect of reference sequence selection on our $\alpha$ estimates, we considered two alignments, one using the *Homo sapiens* genome as reference sequence and the other using the *Mus musculus* genome as reference (*Figure 2—figure supplement 1A*).

In birds, we subdivided the 363-way alignment generated by the B10K project (https://b10k.genomics.cn/) (*Feng et al., 2020*) into six subgroups, avoiding the inclusion of ancestral nodes with high uncertainty within Neoaves (*Feng et al., 2020*; *Jarvis et al., 2014*; *Prum et al., 2015*). Since a species topology is required to accurately infer branch-specific substitution rates, we built species sets by combining monophyletic groups that are well supported across data types and studies (*Reddy et al., 2017*; *Supplementary file 4*). In all cases, we used the *Gallus gallus* genome as the reference sequence.

In snakes, we built our own multiple genome alignments using whole genome assemblies downloaded from the National Center for Biotechnology Information (NCBI) database (*Supplementary file 1*). To speed up computation, we removed repetitive regions – which are ignored in all downstream analyses – from the whole genome FASTA files prior to alignment by converting lowercase bases (i.e., a, t, c, g) to N bases. We ran the Cactus program (v1.2.5, https://github.com/ComparativeGenomicsToolkit/cactus, *Hickey, 2022a*) to align the genomes in each clade using topologies generated by TimeTree as our guide trees (see trees/Snakes.TimeTree.nwk at https://github.com/flw88/mut_sex_bias_amniotes/; *de Manuel, 2022*). For subsequent analyses, we used *Thamnophis elegans* as the reference sequence in snakes.

For each taxon, we converted the HAL file into a Multiple Alignment Format (MAF) file and split the alignment into non-overlapping windows of 1 Mb using the hal2maf tool in halTools (https://github.com/ComparativeGenomicsToolkit/hal/; *Hickey, 2022b*):

```
hal2maf $hal $maf --targetGenomes $species_list --refGenome \
$reference --refSequence $reference_chrom --start $start \
--length $end-$start --onlyOrthologs --noDupes --noAncestors.
```

## Species selection criteria

To estimate $\alpha$, we aimed to measure differences in the rates of neutral substitution in X (Z) versus autosomes that are directly attributable to differences in the mutation rate of males and females. However, X (Z) and autosomes also differ in a number of other technical and biological features that must first be taken into account.

One important source of technical bias is the unequal sequence coverage of the X (Z) and autosomes in heterogametic individuals. To minimize any potential issues due to systematic differences in assembly quality between X (Z) and autosomes, we excluded non-chromosome level genomes known to be assembled exclusively from DNA of the heterogametic sex. In addition, we discarded any species belonging to a genus in which a complex system of chromosomal sex determination has been identified (annotated as 'complex XY' or 'complex ZW' in the Tree of Sex database https://coleoguy.github.io/tos/data.vert.csv, with the exception of the *Mus* genus). In mammals, out of a total of 241 genomes, this approach led us to exclude 50 male-based assemblies and nine species with at least one case of complex XY in the same genus. In birds, out of a total 363 genomes, we excluded 186 female-based assemblies and two species with at least one species with a complex ZW in the same genus.

The quality of the genome assembly is an additional potential confounder. Given that we relied on higher quality, chromosome-level assemblies to categorize alignments as X (Z) or autosomal, we would be more likely to miscategorize alignments (i.e., as X/Z or autosomal) in species with lower quality genome assemblies that are highly diverged from the nearest chromosome-level assembly. To address this issue, in mammals, we removed species if their genomes were >15% diverged from the nearest chromosome-level assembly. We relaxed the divergence threshold to 30% in birds, in which fewer genomes are assembled at chromosome-level and across which karyotypes are believed to be relatively stable (*Ellegren, 2010*). In both mammals and birds, we relied on published divergence estimates inferred from the same multi-alignments used in this study (see mammals *Zoonomia Consortium, 2020* and birds *Feng et al., 2020*). We also discarded species with low-quality scaffold-level assemblies, that is, where scaffold N50 < 350 kb and contig N50 < 25 kb. These filters led to the removal of 120 and 76 species in mammals and birds, respectively.

Given the paucity of genomes in snakes, we relaxed our filtering criteria to allow the inclusion of a larger number of species. Specifically, we allowed scaffold-level assemblies from the heterogametic sex and reduced the scaffold and contig N50 thresholds to 100 and 10 kb, respectively. These changes allowed the inclusion of *Vipera berus* and *Pantherophis obsoletus*. We estimated divergence between species using `phyloFit` (see Estimating putatively neutral substitution rates) in the largest chromosome in *Thamnophis elegans* (NC 045541.1). As in mammals, we removed any species with distance to nearest chromosome-level assembly > 15% and confirmed that none of the species belong to a genus with a complex ZW system in the Tree of Sex database. This procedure excluded one of the nine snake species (*Laticauda laticaudata*).

Another important consideration comes from the differing evolutionary histories of sex chromosomes and autosomes. Under neutrality and assuming equal variances in reproductive success, the X (Z) chromosome is expected to have a lower effective population size, $N_e$, than the autosomes (*Amster and Sella, 2020*). For closely related species, this implies a deeper coalescence time of autosomes than X (Z) in their ancestral population and therefore an unequal contribution of ancestral polymorphisms to the substitution rates; for example, if $N_e^X < N_e^A$, then the X-to-autosome substitution rate ratio will be deflated relative to the expectation under mutational male bias alone, and consequently $\alpha$ will be overestimated (*Presgraves and Yi, 2009*). To minimize this problem, we sought to keep a subset of species that were sufficiently distantly related such that the contribution of ancestral polymorphism to divergence is small and the bias in $\alpha$ estimates is negligible. Specifically, we proceeded as follows: under simplifying assumptions, the expected neutral divergence attributable

to ancestral polymorphisms is given by the heterozygosity, $\pi$, in the ancestral species. Since $\pi$ in the ancestral population of a species pair is unknown, we used estimates for $\pi$ from present-day species as a proxy. We pruned the phylogeny of each taxon so to retain only species pairs with a combined (summed) substitution rate of at least $15\pi$, where $\pi$ is the higher value of the pair.

We collected mammalian $\pi$ estimates from the individual heterozygosities in the Zoonomia Project ('Overall heterozygosity' in Table S3 in ***Zoonomia Consortium, 2020***), complemented with the nucleotide diversities in ***Buffalo, 2021a*** ('log10_diversity' in `data/combined_data.tsv` at https:// github.com/vsbuffalo/paradox_variation/; ***Buffalo, 2021b***) obtaining $\pi$ values for 16 of the remaining mammalian species. For any species lacking a value in both databases, we assigned the $\pi$ of the closest species in the mammalian phylogeny as inferred with PHAST (45 species). Finally, in one case in which $\pi$ from both databases were available (*Daubentonia madagascariensis*), we took the average $\pi$.

In birds, we used $\pi$ estimates in ***Brüniche-Olsen et al., 2021***, obtaining direct estimates for 13 of the remaining species. For species not present in the database, we assigned the $\pi$ of the closest species in the bird phylogeny (85 species). In snakes, we collected $\pi$ values from the literature (***Supplementary file 1***).

Because initiatives like the Zoonomia Project or B10K may preferentially select species at risk of extinction (***Zoonomia Consortium, 2020***), some of the present-day $\pi$ values may underestimate the diversity levels in the ancestor. We thus set an extra requirement of at least a combined 2% substitution rate between any pair of species. In species pairs where the rate was below either of these two thresholds ($15\pi$ or 2%), we preferentially retained the species that met the following criteria, considered in this order: (1) more phased DNM count data from pedigree sequencing (count of 0 if not available), (2) a chromosome-level assembly, and (3) a higher scaffold N50. Altogether, 20 out of 241 mammalian species, 17 out of 363 bird species, and 5 out of 9 snakes species remained after the complete filtering procedure.

A list of the species kept after filtering, together with other genome statistics and results from our analyses, can be found in ***Supplementary file 2***. The code to reproduce the filtering procedure described above can be found in `notebooks/Filter_species.ipynb` at https://github.com/ flw88/mut_sex_bias_amniotes; ***de Manuel, 2022***.

## Selecting non-repetitive and putatively neutral sequences

In the absence of natural selection and/or gBGC, the substitution rate is equal to the mutation rate (***Kimura, 1983***). To minimize the effects of selection, we limited our analyses to non-coding regions by removing all exons annotated in the given reference sequence as well as the 1 kb of sequence flanking each exon. As a check, we also estimated $\alpha$ in mammals and birds after masking conserved elements identified by phastCons (***Siepel et al., 2005***) (mammals and birds, respectively). Since the $\hat{\alpha}_{evo}$ are similar, we based our analyses on the larger dataset based on masking only exons and their 1 kb flanking sequences.

The effect of gBGC mutation on the substitution process is analogous to that of selection for specific base pairs, in that the process increases the probability of fixation of strong (G/C) over weak (A/T) alleles (***Duret and Galtier, 2009***). To explore the effects of gBGC, we estimated specific rates for each single-nucleotide substitution type (see Estimating putatively neutral substitution rates for details). To remove the effects of gBGC, we estimated $\alpha$ for the subset of mutation types that are not subject to gBGC (i.e., substitutions from strong to strong and weak to weak nucleotides) (***Figure 2— figure supplement 3*** and ***Figure 2—figure supplement 1G***).

In addition, to ensure the high quality of the alignment data for analysis, we removed repetitive regions, keeping only those genomic positions at which the reference sequence in a given analysis group (mammals, birds, and snakes) carries an uppercase nucleotide.

## Filtering idiosyncratic genomic regions

We excluded sequences aligned to known PARs in the sex chromosomes, which have homologs on both X and Y (or both Z and W) and thus behave like autosomes in terms of their ploidy (see ***Supplementary file 5*** for PAR definitions). For snakes, we aligned sequenced reads from a female *Thamnophis sirtalis* individual (NCBI accession SAMN02402779) to the *Thamnophis elegans* reference genome using BWA-MEM v0.7.17-r1188 (http://bio-bwa.sourceforge.net/), with default parameters. We removed PCR duplicates with the markdup tool in samtools v1.10 (http://www.htslib.org/) and

calculated the mean depth of coverage along the Z and the largest autosome in 1 Mb windows using *mosdepth* (https://github.com/brentp/mosdepth, *Pedersen, 2022*). We then determined regions of the Z chromosome in which the depth of coverage was significantly different to that in the autosomes, assuming depth is Poisson distributed with $\lambda$ equal to the mean depth in the autosome, potentially indicative of the region being in a PAR and having homologs on the W chromosome (*Figure 1—figure supplement 1*).

The genome of birds and snakes are organized into two types of autosomes, macro- and micro-chromosomes, which differ in their length, gene content, density of hypomethylated CpG islands, recombination rates, and replication timing (*Waters et al., 2021*). Given the idiosyncrasies of micro-chromosomes, which may affect the substitution rate estimates (*Wang et al., 2014*), we excluded sequences aligned to microchromosomes in birds and snakes (chromosomes 10–28 in *Gallus gallus* and chromosomes 13–18 in *Crotalus viridis*). The fraction of base pairs in microchromosomes is relatively small, comprising 20% and 5.1% of the autosomal genome in *Gallus* and *Crotalus*, respectively. We checked that $\hat{\alpha}_{evo}$ are similar whether or not microchromosomes are excluded ($r > 0.9$ between $\hat{\alpha}_{evo}$ estimates obtained after excluding or including microchromosomes, in both birds and snakes, *Figure 2—figure supplement 1E*).

An additional concern is that genomic translocations between X (Z) and the autosomes could lead to sequence misclassification in species without a chromosome-level assembly. To alleviate this potential issue, we only kept sequences that exclusively mapped to chromosomes of the same kind (i.e., X or Z versus autosome) in all species for which chromosome-level assemblies were available. In other words, we removed all alignments in which chromosome-level assemblies indicated a mapping between an X (Z) sequence of one species with an autosomal or Y (W) sequence of another.

To summarize, each 1 Mb MAF file in each taxon was first filtered with the maf_parse tool in PHAST (http://compgen.cshl.edu/phast/), using a thinned set of species obtained as described in Species selection criteria and a BED file with the regions to be excluded as indicated by the reference genome (i.e., exons ±1 kb and the PARs, if known). The python scripts `filter_PARs_micros_CpGs.py` and `keep_species_XYA-synteny.py` (available at https://github.com/flw88/mut_sex_bias_amniotes, copy archived at swh:1:rev:37da9bdbc2c7cb839de15aadb554cf6c98128add; *de Manuel, 2022*) were then used to filter any gaps, annotated PARs, as well as regions that mapped to known chromosomes of a different kind:

```
maf_parse --features $regions_to_exclude_bed -M $reference \
--seqs $(cat $species_list_thinned) $maf |
python filter_PARs_micros_CpGs -p data/Species_to_PARs.tsv |
python filter_species_gaps_maf_XYA.py \
-l $species_list_thinned -c data/Species_to_chromosomes.txt \
-b $filtered_regions_bed -a > $filtered_maf
```

## GC content and replication timing estimates

The framework provided by *Miyata et al., 1987*, to infer $\alpha$ assumes that the generation time is the same for both sexes, as well as that the substitution rates on autosomes versus X (Z) are solely determined by the sex-specific mutation rates and the ploidy difference between sexes. However, other genomic features, such as GC content and replication timing, are known to differentially influence the mutation rate of sex-linked and autosomal chromosomes (*Agarwal and Przeworski, 2019*; *Koren et al., 2012*). To account for these differences, we collected measures of species-specific GC content. Specifically, for every filtered 1 Mb MAF in each taxon, we calculated the fraction of G/C base pairs in each genome with:

```
cat $filtered_maf | \
python gc_content_from_maf.py -s $species_list_thinned
```

We additionally obtained replication timing data in human embryonic stem cells from the UCSC genome browser. We converted the data from bigWig format to BED using bigWigToBedGraph and lifted the coordinates from the hg19 reference genome to hg38 using the liftOver tool.

To explore the relationship between replication timing and substitution rates in humans, we calculated an average replication timing value across the unfiltered bases in each 1 Mb window of the mammalian alignment (*H. sapiens* as reference). Specifically, we used the mean replication timing value weighted by the number of bases associated with each replication timing datum.

## Estimating putatively neutral substitution rates

To estimate putatively neutral substitution rates on X (Z) and autosomes, we used `phyloFit` a program within the PHAST software suite (*Hubisz et al., 2011*; *Siepel and Haussler, 2004*) (http://compgen.cshl.edu/phast/). For every 1 Mb window of aligned sequence in each taxon with ≥10 kb of sequence remaining after filtering, we estimated substitution rates using the general, unrestricted single nucleotide model (`--subst-mod UNREST`) with the expectation maximization algorithm with medium precision for convergence (`--EM --precision MED`). We also obtained the number of expected counts at each node for each substitution type (option `-Z`). For mammals and birds, we used the relevant tree topology defined in the Newick files in http://cgl.gi.ucsc.edu/data/cactus/; for snakes, we used a topology from TimeTree (http://timetree.org/). To avoid local maxima in the likelihood surface, we ran six independent `phyloFit` runs with random initialization of the parameters (option `-r`) and kept the replicate with the highest likelihood. We note that phyloFit estimates the expected substitution counts for type $A_1 > A_2$ by inferring the expected number of times allele $A_1$ is found at the internal node of a branch in the tree and allele $A_2$ is observed at the terminal node. However, the overall branch lengths are maximum likelihood estimates of the expected rate of substitution in continuous time along the branches. Thus, the rate of substitution estimated by summing substitution counts and dividing by the genome size is slightly smaller than the maximum likelihood branch-length estimate (as the latter allows back-mutation but the former does not include them).

```
phyloFit -r --EM --precision MED --subst-mod UNREST -Z \
--msa-format MAF $filtered_maf --tree $newick \
-e $phylofit_errors -o $phylofit_output.
```

## Estimating $\alpha$ from X-to-autosome substitution rate ratios

We took a regression approach to estimate $\alpha$ from ratios of X (Z)-to-autosome substitution rates. This approach allowed us to control for the effect of GC content $g$ on the substitution rates (see Estimating putatively neutral substitution rates). For each species, we performed a Poisson regression with a log link function on the number of substitutions $Y_i$ in the terminal branch (as inferred from `phyloFit`):

$$\log\left[E\left(Y_i\middle|n_i, x_i, g_i\right)\right] = \log(n_i) + \beta_0 + \beta_1 x_i + \beta_2 g_i + \beta_3 g_i^2 \tag{3}$$

where the subscript denotes the ith window, $n$ denotes the number of bases at which a substitution could have occurred, $x$ is an indicator variable denoting whether the window is on the X (Z) or the autosomes, and the $\beta$ variables denote the regression coefficients (*Supplementary file 6*). Modeling the relationship between substitution rate and GC content as a quadratic function captures effects of hypermutable CpG sites via the squared term (*Hardison et al., 2003*; *Hellmann et al., 2005*). Note that for the overall substitution count, the number of substitution opportunities $n$ is the total number of sites left in the window after filtering; however, when applying the regression model to a specific substitution type $A_1 > A_2$, we only considered sites where the ancestral allele was inferred by `phyloFit` to be $A_1$ (or its complementary base, see Estimating putatively neutral substitution rates).

We used the fitted regression models to estimate $\alpha$ in each species. To this end, we first obtained point estimates of the substitution rates on the X (Z) and autosomes calculated at the mean GC content values of the X (Z) windows. We then converted the resulting X (Z)-to-autosome substitution rate ratio to an estimate of $\alpha$ using Miyata's equations (*Miyata et al., 1987*). This approach implicitly assumes that mutation rates in X (Z) and autosomes differ only with regard to their exposure to sex, once differences in pertinent genomic features are taken into account.

We note that this approach infers $\alpha$ from the ratio of the expectations of the X (Z) and autosomal substitution rates rather than the expectation of the ratios. To check whether that makes a difference, we re-estimated $\alpha$ in each species using a modified procedure in which we repeatedly sampled a pair of X (Z) and autosome windows with GC content values in a narrow range (mean GC content value of the X (Z) chromosome $\pm1.5\%$) and calculated a X (Z)-autosome substitution rate ratio. Estimating $\alpha$ from the mean ratio across 1000 resamples yielded highly similar estimates to those obtained from our regression approach ($r = 0.93$ across species, *Figure 2—figure supplement 1*).

To understand whether controlling for replication timing in addition to GC content might affect our $\alpha$ estimates, we modified *Equation 3* to include an extra term for the average replication timing of each window $t_i$ (see GC content and replication timing estimates). We applied this modified regression framework to mammals and obtained X-autosome substitution rate ratios for each species at the

mean GC content and replication timing values of the X windows. Converting the X-autosome substitution rate ratios to $\alpha$ estimates using Miyata's equations (*Miyata et al., 1987*) yielded values that were highly similar to those obtained when controlling for GC content only ($r > 0.99$, see *Figure 2—figure supplement 1B*). Given the observed agreement and the lack of replication timing data for most species, in subsequent analyses, we relied on evolutionary estimates obtained from the regression model described in *Equation 3*.

To assess the uncertainty in our $\alpha$ estimates, we bootstrap resampled windows on the X (Z) and autosomes 500 times. For each replicate, we fit the regression model and calculated the X (Z)-to-autosome ratio as described above to obtain an empirical distribution from which we could compute the central 95% interval. We note that because of the functional form describing the relationship between $\alpha$ and the X (Z)-to-autosome substitution rate ratio (*Figure 1A*), confidence intervals on $\alpha$ tend to be wider at larger values of $\alpha$. In other words, in the regime of large $\alpha$, a small shift in the X (Z)-to-autosome substitution rate ratio will have a larger impact on the inferred $\alpha$ estimate. We implemented our regression and $\alpha$ estimation framework in the R script, `alpha_from_unrest.regression.R`.

Although ignored in the original Miyata et al. approach and subsequent applications (e.g., *Wilson Sayres et al., 2011*; *Wang et al., 2014*; *Schield et al., 2021*), recent modeling work shows that sex differences in generation times can also affect the relative ratio of substitution rates on the X (Z) and autosome by altering the amount of time that a sex chromosome lineage spends in males versus females compared to autosomes (*Amster and Sella, 2016*). Thus, sex differences in generation times modulate how sex biases in mutations are reflected in substitution rates of X (Z) versus autosomes. Unfortunately, sex-specific generation time estimates are rarely available for extant species, let alone ancestral lineages, and likely evolve over time. To incorporate uncertainty in sex differences in generation times, we re-computed our uncertainty intervals on $\alpha$ under the assumption that the male-to-female ratio of the generation times for any particular lineage lies between 0.9 and 1.1, using formulas derived by *Amster and Sella, 2016*.

## Estimating $\alpha$ from pedigree studies in vertebrates

In order to obtain estimates of $\alpha$ from extant vertebrate species, we identified 14 DNM studies with published counts of parentally phased DNMs (*Bergeron et al., 2021*; *Besenbacher et al., 2019*; *Campbell et al., 2021*; *Harland et al., 2017*; *Jónsson et al., 2017*; *Lindsay et al., 2019*; *Smeds et al., 2016*; *Tatsumoto et al., 2017*; *Thomas et al., 2018*; *Wang et al., 2020*; *Wang et al., 2022a*; *Wang et al., 2022b*; *Wu et al., 2020*; *Yang et al., 2021*). For each species in each study, we calculated point estimates of $\alpha$ by dividing the number of DNMs phased to the paternal chromosome by the number phased to the maternal chromosome (*Supplementary file 2*). We measured uncertainty by computing binomial confidence intervals on the proportion of all phased DNMs that were paternal and then converting the resulting interval bounds back to a paternal-to-maternal ratio.

From this list, we excluded one study from mouse lemur (*Microcebus murinus*), which reported an anomalously high mutation rate per year for a primate species ($> 3.5 \times 10^{-9}$ per site) and unusually low rates of transitions at CpG sites (*Campbell et al., 2021*). The authors suggested C-to-T substitutions in the branch leading to mouse lemur occurred at a similar rate irrespective of their dinucleotide context (CpG or non-CpG), in contrast to what is seen in other primates (*Moorjani et al., 2016*). However, analyzing our substitution data, we find the C>T substitution rate in mouse lemur to be over fivefold higher at CpG sites compared to non-CpG sites. Specifically, we estimated substitution rates from our filtered autosomal mammalian alignments as described in Estimating putatively neutral substitution rates with the following modifications: (I) CpG islands, as defined here, were masked following *Campbell et al., 2021*; (II) CpG dinucleotide substitution rates were estimated using a context-dependent model (`--subst-mod U2S`). This study also reports the weakest mammalian paternal bias in mutation described to date ($\alpha = 1.18$). This value is out of sync with reports for other primates and far from what we estimate from substitution rates, $\hat{\alpha}_{evo}$ (*Figure 2* and *Supplementary file 2*). One possibility is that a substantial rate of false positive DNMs biased $\hat{\alpha}_{dnm}$ toward 1 (since errors are likely placed with equal probability on the maternal or paternal haplotype). Given the uncertainty surrounding how to interpret these DNM data, we do not include this $\hat{\alpha}_{dnm}$ in our analyses.

## Estimating $\alpha$ for different developmental stages

DNM studies typically quantify the number of mutations in the offspring that are not found in some somatic tissue (usually blood) of the parents. This approach can mistakenly include DNMs that occurred in the early development of the offspring, as well as mistakenly exclude DNMs that occurred early in the development of the parents (*Gao et al., 2016*). DNMs that occurred in early development of the parents can be distinguished by patterns of 'incomplete linkage' with nearby informative constitutive heterozygous positions, as well as incomplete transmission to the offspring (*Harland et al., 2017*; *Sasani et al., 2019*). Moreover, DNMs that occurred right after or during primordial germ cell specification (PGCS) will not be present in the soma of the parents but may be transmitted to multiple offspring (*Sasani et al., 2019*; *Lindsay et al., 2019*).

To examine if $\alpha$ varies across developmental stages, we considered studies that distinguish between DNMs in the early development of the parent (i.e., mutations detectable in the parental soma but showing patterns of 'incomplete linkage', as well as DNMs transmitted to multiple offspring), versus DNMs that occurred in later stages after PGCS (i.e., not present in the parental soma and transmitted to a single offspring). Counts for early DNMs were obtained: in mice (*Lindsay et al., 2019*), where we counted the number of mutations phased to each parental haplotype in 'Early Embryonic' and 'Peri-PGC' categories (Supplementary Data 1 at https://doi.org/10.1038/s41467-019-12023-w); in cattle (*Harland et al., 2017*), where we counted mutations classified as 'Sire Mosaic' or 'Dam Mosaic' (Supplementary Table 1 at https://doi.org/10.1101/079863); and in humans (*Sasani et al., 2019*), where we counted the number of mutations phased to each parental haplotype in 'Gonosomal mutations' and 'Post-PGCS' (Tables in https://github.com/quinlan-lab/ceph-dnm-manuscript/tree/master/data). DNM counts for phases later in development were obtained from the same publications, under the categories 'Late post-PGCS', 'Sire/Dam non Mosaic', and 'Third-generation' in mice, cattle, and humans, respectively. All three studies also employed strategies to discard DNMs in the early development of the offspring. The combined counts for each species and mutation timings can be found in *Supplementary file 3*.

Since the paternal bias in mutation varies among developmental stages, as does the fraction of mutations that were successfully phased (*Supplementary file 3*), simply summing over DNM counts from different stages would result in a biased point estimates of the overall $\alpha$. We therefore computed $\alpha$ by extrapolating the proportion of paternally and maternally phased DNMs in each stage to all the DNMs identified in that stage (i.e., extrapolating to what would be expected given complete phasing). Given this extrapolation, the measures of uncertainty associated with 'Total' are not shown in *Figure 4A*. For DNMs within a single developmental stage, we calculated binomial confidence intervals, as described above.

## Testing relationships between $\alpha$ and life history traits

In mammals, we collected life history traits from the AnAge database (https://genomics.senescence.info/species/dataset.zip), including maximum longevity, gestation time, adult weight, and birth weight. We also obtained generation time estimates from the literature (*Supplementary file 2*). Thus, in total, we collected data on five traits. Four species were not represented in the AnAge dataset; in these cases, we substituted the trait values of closely related species of the same genus (see *Supplementary file 2* for species substitutions). We additionally performed principal component analysis (PCA) on the four traits, generation time, gestation time, adult weight, and birth weight (*Figure 3—figure supplement 2*), and treated PC1 and PC2 as meta-traits to be tested alongside the others. Only the 17 mammalian species annotated for all four traits were included in the PCA procedure. The first principal component captured 90% of the variance in the traits and was highly correlated with generation time ($r^2 = 86\%$). In birds, we focused on the life history trait of generation time, taking estimates from the literature (*Supplementary file 2*).

To test for relationships between life history traits and $\alpha$ while accounting for phylogenetic non-independence in our data, we used phylogenetic generalized least squares (PGLS) (*Grafen, 1989*). Ordinary least squares is unsuitable for species trait comparisons, because shared phylogenetic history can create correlation structure in the residuals (*Felsenstein, 1985*). PGLS addresses this issue by considering the covariance structure of the residuals as a covariate, assuming that the traits evolve under Brownian motion on the phylogeny (*Grafen, 1989*; *Pagel, 1999*). We implemented the analysis using the `pgls` function in the caper R package, which provides the option of fitting Pagel's $\lambda$

(*Pagel, 1999*), a scalar multiplier of the off-diagonal elements of the expected covariance matrix of the residuals. Briefly, $\lambda$ denotes the amount of phylogenetic 'signal' in the data. If $\lambda$ is 0, there is no phylogenetic signal; when $\lambda$ is 1, the regression model is equivalent to the method of phylogenetic independent contrasts (PIC) (*Blomberg et al., 2012*; *Felsenstein, 1985*; *Pagel, 1999*). In practice, we found that the `pgls` R function would occasionally fail to converge or converge on a local maximum during maximum likelihood estimation of $\lambda$; to address this issue, we initialized the likelihood optimization algorithm with a variety of starting values for $\lambda$ and retained the model with the highest overall likelihood, which required a minor modification of the base `pgls` function from the caper package.

For each predictive trait (*Figure 3—figure supplement 1*), we used our $\hat{\alpha}_{\text{evo}}$ estimates from X (Z)-to-autosome comparisons as the response variable and a time-calibrated phylogeny from Time-Tree to estimate the covariance matrix (http://timetree.org/). Following what had been done previously to analyze these relationships (*Wilson Sayres et al., 2011*), we log10-transformed each life history trait prior to performing PGLS. *Canis lupus familiaris*, *Ceratotherium simum cottoni*, and *Pterocles burchelli* were not named in the TimeTree database and so we used split times for *Canis lupus*, *Ceratotherium simum*, and *Pterocles gutturalis* instead, respectively (*Supplementary file 2*). In all comparisons, we calculated p-values under a model in which $\lambda$ was set to its maximum likelihood estimate and used default values for the remaining arguments of the `pgls` program. In birds, in which the MLE for $\lambda$ was 0, we also considered a model in which $\lambda$ was fixed at 1. To test whether the slope of the $\hat{\alpha}_{\text{evo}}$ versus generation time relationship is the same in birds as in mammals, we performed a modified PGLS regression on the bird data with the slope fixed to the maximum likelihood value obtained for mammals (i.e., slope = 1.20) and the intercept (and $\lambda$) as the free parameter. After fitting this model with PGLS, we performed a likelihood ratio test (df = 1) to compare it to an alternative model in which the slope was not fixed (i.e., including intercept, slope, and $\lambda$ parameters).

## Modeling the effects of germline developmental stages on $\alpha$

To model variation in $\alpha$ among species, we considered the expected number of mutations that arise in two developmental stages: an early embryonic period, *Early*, which loosely encompasses the time between the zygote and the sexual differentiation of the germline, and a second period, *Late*, that refers to the remaining time until reproduction (*Figure 4B*). In mammals, the expected number of mutations in the *Early* stage, $M_e$, is approximately the same in both sexes, as observed in the three cases in which there are data (*Figure 4A*). In the *Late* stage, we assume mutations arise at a constant rate per year, $\mu_s$ in sex $s$ ($s \in \{f, m\}$). If we assume the length of *Early* to be negligible relative to the generation time, $G_s$ in sex $s$, then the expected number of mutations in sex $s$ equals $\mu_s G_s$. Therefore, the expectation of the ratio of paternal-to-maternal mutations at reproduction, $\alpha$, can be obtained using *Equation 1*.

To predict $\alpha$ in species lacking estimates of the sex-specific mutation rates for the *Late* stage (i.e., $\mu_m$ and $\mu_f$), we made two further assumptions, namely that:

- The expected number of mutations per base pair $M_e$ in the *Early* stage is constant across species and the same in the two sexes. We used an $M_e$ of $1.66 \times 10^{-9}$ per base pair, which equates to five early embryonic mutations in an haploid genome of 3 Gb. This value was chosen based on observations in humans, notably a study showing that monozygotic twins differ on average by 5.2 mutations that arose between the twinning event and PGCS (1.3 mutations per haploid set of chromosomes) (*Jonsson et al., 2021*). Given that 75–80% twinning events occur around the 8–16 cell stage (*Hall, 2003*), approximately four mutations are expected to have arisen during the first few divisions in the embryo (assuming ~1 extra mutation per cell division; *Ju et al., 2017*). This rate is also in rough agreement with a pedigree study in humans, which estimated that ~5% of DNMs arise during early development (*Sasani et al., 2019*). Varying the expected number from 3 to 7 yielded similar results (see below for more details).
- The ratio $\mu_m/\mu_f$ is fixed across species. We assumed a ratio of 4, consistent with the ratio of paternal-to-maternal DNMs occurring post-PGCS in humans (*Sasani et al., 2019*), mice (*Lindsay et al., 2019*), and cattle (*Harland et al., 2017*) (*Figure 4A*).

Using derivations from *Amster and Sella, 2016*, the yearly substitution rate $\mu_Y$ for a given lineage is:

$$\mu_Y = \frac{2M_e + \mu_f G_f + \mu_m G_m}{G_f + G_m}.$$

If $\mu_m/\mu_f = 4$ and $M_e$ is known, we can solve for $\mu_f$ using **Equation 2** and $\alpha$ can be estimated using **Equation 1**.

We used the PGLS method described in 'Testing relationships between $\alpha$ and life history traits' to assess the fit of $\alpha$ values predicted by our model to the $\alpha$ values estimated from X-to-autosome comparisons ($\hat{\alpha}_{evo}$) and from DNM studies ($\hat{\alpha}_{dnm}$) (**Figure 4C**). We applied the model to mammals using estimates of $G$ from the literature (**Supplementary file 2**). When testing the fit of the model to $\hat{\alpha}_{evo}$, we estimated $\mu_Y$ by dividing the autosomal substitution rates in a lineage (see Estimating putatively neutral substitution rates) by the split time for that lineage reported in the TimeTree database (http://timetree.org/). When testing the fit to $\hat{\alpha}_{dnm}$, we obtained $\alpha$ from yearly mutation rates obtained from pedigree sequencing studies, given the parental ages in the study (see **Supplementary file 2**). We note that $\hat{\alpha}_{dnm}$ can be noisy if not based on a large amount of DNMs and trios. To overcome this limitation, we focused on species with at least 30 phased DNMs and more than one trio sequenced (which excluded three species out of 14, namely *Pongo abelii*, *Callithrix jacchus*, and *Ursus arctos*, see **Supplementary file 2**).

We note the model remains a significant predictor for a range of $M_e$ values. As examples, using a $\lambda$ of 1, as inferred by maximum likelihood in **Figure 4D**, for an $M_e = 1 \times 10^{-9}$, the model for $\hat{\alpha}_{evo}$ explains $r^2 = 0.33$ (p-value = 0.008) and for $\hat{\alpha}_{dnm}$, $r^2 = 0.90$ (p-value = $1 \times 10^{-4}$). Instead using $M_e = 2.33 \times 10^{-9}$, the model for $\hat{\alpha}_{evo}$ accounts for $r^2 = 0.35$ (p-value = 0.006) and for $\hat{\alpha}_{dnm}$, $r^2 = 0.79$ (p-value = 0.001).

Following (**Kong et al., 2012**), we sought to determine the extent to which variation in $\hat{\alpha}_{evo}$ in mammals is attributable to sampling error. To that end, we made use of the empirical distribution of $\hat{\alpha}_{evo}$, which we obtained by bootstrap resampling genomic windows (see Estimating $\alpha$ from X-to-autosome substitution rate ratios). For each bootstrap replicate, we regressed the $\alpha$ estimates against our original $\hat{\alpha}_{evo}$ using ordinary least squares and obtained the $r^2$ value. Across the 500 bootstrap replicates, the median $r^2$ value was 89%, suggesting that 11% of the variance in $\hat{\alpha}_{evo}$ is due to sampling error. Combining this value with the estimated proportion of variance in $\hat{\alpha}_{evo}$ explained by our model yielded an estimate of 37%/89% = 42% of the variance explained after accounting for sampling error.

The code to reproduce the modelling described above can be found in the `scripts/2exposure_model.ipynb` Jupyter notebook.

## Acknowledgements

We thank Ziyue Gao, Guy Sella, and the Coop and Schierup labs for their comments on earlier versions of the manuscript. We thank Rusty Lansford, Mike McGrew, and Daniel Hooper for discussions about avian development and evolution; Turk Rhen for discussions about reptile sex determination; Carla Hoge and Zach Fuller for sharing their corn snake genome assembly; Anne Bronikowski and the Vertebrate Genome Project for sponsoring and generating the *Thamnophis elegans* assembly; Alex Cagan for early access to data of mutation burdens in colonic crypts across mammals; Richard Wang and Matthew Hahn for sharing data on DNM in cats; Carole Charlier and Michel Georges for sharing data on DNM in cattle; Adam Siepel for help with applying the phyloFit program; and Peter Andolfatto, Michael B Eisen, Priya Moorjani, as well as William R Milligan, Anna Yoney, and other members of the Andolfatto, Przeworski, and Sella labs for helpful discussions. This work was funded by GM122975 to MP and an HFSP postdoctoral fellowship to MdM.

## Additional information

### Competing interests

Molly Przeworski: Senior editor, *eLife*. The other authors declare that no competing interests exist.

### Funding

| Funder | Grant reference number | Author |
| --- | --- | --- |
| National Institutes of Health | GM122975 | Molly Przeworski |

| Funder | Grant reference number | Author |
|---|---|---|
| Human Frontier Science Program | LT000257/2021-L | Marc de Manuel |

The funders had no role in study design, data collection and interpretation, or the decision to submit the work for publication.

## Author contributions

Marc de Manuel, Felix L Wu, Data curation, Formal analysis, Investigation, Visualization, Methodology, Writing – original draft, Writing – review and editing; Molly Przeworski, Conceptualization, Supervision, Funding acquisition, Writing – original draft, Project administration, Writing – review and editing

## Author ORCIDs

Marc de Manuel http://orcid.org/0000-0002-1245-0127
Felix L Wu http://orcid.org/0000-0002-0155-9071
Molly Przeworski http://orcid.org/0000-0002-5369-9009

## Decision letter and Author response

Decision letter https://doi.org/10.7554/eLife.80008.sa1
Author response https://doi.org/10.7554/eLife.80008.sa2

# Additional files

## Supplementary files

• Supplementary file 1. Genome assembly statistics and heterozygosity estimates for 241 mammals, 365 birds, and 9 snakes.

• Supplementary file 2. Estimates of $\alpha$ obtained from the ratios of X (Z)-autosome substitution rates and from pedigree data, as well as life history traits, and genome assembly statistics for 46 mammal, bird, and snake species.

• Supplementary file 3. Phased de novo mutation counts categorized by developmental stage.

• Supplementary file 4. Sets of bird species used to estimate $\hat{\alpha}_{evo}$. To avoid uncertain phylogenetic relationships within Neoaves, birds were split into separate analysis sets before estimating substitution rates (see Sequence alignments in Materials and methods). In groups 1–5, *Gallus gallus* was included as an outgroup when estimating substitution rates.

• Supplementary file 5. Pseudo-autosomal region intervals. Coordinates were taken from studies *Liu et al., 2019*; *Raudsepp and Chowdhary, 2015*; *Schield et al., 2019*; *Shearn et al., 2020*; *Skinner et al., 2013*; *Das et al., 2009*; *Smeds et al., 2014*.

• Supplementary file 6. Parameter estimates and 95% confidence intervals from the regression of substitution rates against GC content (see *Equation 3*).

• MDAR checklist

## Data availability

All source data and scripts to reproduce the findings in the manuscript can be found at https://github.com/flw88/mut_sex_bias_amniotes, (copy archived at swh:1:rev:37da9bdbc2c7cb839de15aadb554cf6c98128add).

The following previously published datasets were used:

| Author(s) | Year | Dataset title | Dataset URL | Database and Identifier |
|---|---|---|---|---|
| Armstrong J, Hickey G, Diekhans M, Fiddes IT, Noval AM, Deran A, Fang Q, Xie D, Feng S, Stiller J, Grenereux D, Johnson J, Marinescu VD, Alföldi J, Harris RS, Lindblad-Toh K, Haussler D, Karlsson E, Jarvis ED, Zhang G, Paten B | 2020 | Mammal and bird whole-genome alignment | https://cglgenomics.ucsc.edu/data/cactus/ | UC Santa Cruz Computational Genomics Lab & Platform, /data/cactus/ |
| Sasani T, Quinlan A | 2019 | Human de novo mutations | https://github.com/quinlan-lab/ceph-dnm-manuscript | github, quinlan-lab/ceph-dnm-manuscript |

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
