## [Editor Report]

This paper challenges a fundamental view concerning why males of most animals have a higher germline mutation rate than females. Evidence is provided to show that it is not simply the fact that males have more cell divisions in the germline, but instead, most of the mutations arise from a different balance of DNA damage vs. DNA repair. The case is supported by data from multiple species, from de novo mutation rate estimates from pedigrees, and from fits to a simple heuristic model. This work will be of interest to the broad field of DNA mutations and DNA repair, as well as evolutionary and phylogenomics researchers.

---

## [Decision Letter]

**Decision letter after peer review:**

Thank you for submitting your article "A paternal bias in germline mutation is widespread in amniotes and can arise independently of cell divisions" for consideration by *eLife*. Your article has been reviewed by 3 peer reviewers, including Yukiko M Yamashita as Reviewing Editor and Reviewer #1, and the evaluation has been overseen by Christian Landry as the Senior Editor. The following individual involved in the review of your submission has agreed to reveal their identity: Andrew G Clark (Reviewer #2).

As you can see in individual reviews, reviewers are overall positive on this manuscript. Most of the comments can be addressed by textual changes for clarification. Please provide point-by-point responses for their comments, and where you feel changes are unnecessary, please provide the rationale for your sentiment.

*Reviewer #1 (Recommendations for the authors):*

– Line 145: authors mention that a(evo) and α (dnm) are very similar, but there is no actual data or comparison presented. It would be helpful if they can provide the figure/table to compare their α (evo) with known α (dnm) data, with references to the data source.

*Reviewer #2 (Recommendations for the authors):*

1. Alpha was estimated for birds with and without the inclusion of microchromosomes and they were found to be correlated. The authors were conservative and mostly report results with the microchromosomes excluded. But since microchromosomes are more gene dense, I am curious how alpha for the micro- and macrochromosomes compare. Reporting both in the supplement would be of interest to many readers.

2. In Section 4.7 (Methods) the authors were careful to consider attributes like GC content and replication timing as additional factors that are known to impact mutation rate. Please report all the beta estimates from Equation 3. We might get worried if the effect of GC content varied wildly across species.

*Reviewer #3 (Recommendations for the authors):*

1. How many epigenetic modifications such as methylation and or imprinting on chrX might bias alpha value?

2. For the mammals the authors have replication timing estimates for could they run on the late replicating and early replicating regions to see if this biases results?

3. It would be interesting to compare mutational signatures per chrX vs. autosomes per species and explore if there are significant variations in the contribution of certain signatures? The result might also give a clearer explanation of cellular processes that causes paternal bias.

4. It is interesting that the alpha values in cats and dogs are quite different. Can this be explained by variation in PRDM9 activity? E.g., subtle differences in recombination rate in females compared to males. It is suggested that most recombination in male dogs is centred around CpG sites while in females the recombination sites are spread more homogeneously across the chromosome.

---

## [Author Response]

Reviewer #1 (Recommendations for the authors):– Line 145: authors mention that a(evo) and α (dnm) are very similar, but there is no actual data or comparison presented. It would be helpful if they can provide the figure/table to compare their α (evo) with known α (dnm) data, with references to the data source.

We had included a Table with these data (Table S2), in addition to presenting the results in Figure 1 (see the legend).

Reviewer #2 (Recommendations for the authors):1. Alpha was estimated for birds with and without the inclusion of microchromosomes and they were found to be correlated. The authors were conservative and mostly report results with the microchromosomes excluded. But since microchromosomes are more gene dense, I am curious how alpha for the micro- and macrochromosomes compare. Reporting both in the supplement would be of interest to many readers.

We had included a figure showing this difference (panel I in Figure S1).

2. In Section 4.7 (Methods) the authors were careful to consider attributes like GC content and replication timing as additional factors that are known to impact mutation rate. Please report all the beta estimates from Equation 3. We might get worried if the effect of GC content varied wildly across species.

We have now added a supplementary table (Table S6).

Reviewer #3 (Recommendations for the authors):1. How many epigenetic modifications such as methylation and or imprinting on chrX might bias alpha value?

As noted in response to reviewer 2, in the germline, the X chromosome is only transiently imprinted in mice/humans (Chuva de Sousa Lopes et al., 2008; Guo et al., 2015), so we do not expect much of an impact on mutation rates on X versus autosomes. With regard to differences in DNA methylation patterns, excluding CpG sites from our analysis does not change the qualitative conclusions (see Figure S1). We now mention X-inactivation explicitly (lines 139-143) and comment that our regression model is highly unlikely to control for all factors other than exposure to sex, in ways that could affect our quantitative estimates (lines 165-167, also see our response to the first point of reviewer #2).

2. For the mammals the authors have replication timing estimates for could they run on the late replicating and early replicating regions to see if this biases results?

In Figure S1B, we showed the results controlling for replication timing; there is very little effect on our conclusions.

3. It would be interesting to compare mutational signatures per chrX vs. autosomes per species and explore if there are significant variations in the contribution of certain signatures? The result might also give a clearer explanation of cellular processes that causes paternal bias.

We agree that such an analysis would be of interest and we had tried to implement it, but substitution patterns are unfortunately confounded by GC-biased gene conversion, and suitable de novo mutation data are lacking. As noted in response to reviewer 1, we are now analyzing a set of pedigrees from birds and reptiles in the hope of addressing this question, but this project is beyond the scope of this paper.

4. It is interesting that the α values in cats and dogs are quite different. Can this be explained by variation in PRDM9 activity? E.g., subtle differences in recombination rate in females compared to males. It is suggested that most recombination in male dogs is centred around CpG sites while in females the recombination sites are spread more homogeneously across the chromosome.

To clarify, the true α of cats and dogs is likely not different from that of other mammals, given that the one pedigree study in cats suggests a paternal bias of 2-3. Instead it is our estimate of α based on substitution data that appears to be biased downwards, for reasons we do not understand (as noted lines 169-175). We are unclear as to why sex differences in recombination would affect our estimate; however, as mentioned (lines 169-175), the X chromosome of cats appears to contain unusual features that might be biasing our estimate, and could be related to recombination.

References:

Chuva de Sousa Lopes, Susana M., Katsuhiko Hayashi, Tanya C. Shovlin, Will Mifsud, M. Azim Surani, and Anne McLaren. 2008. “X Chromosome Activity in Mouse XX Primordial Germ Cells.” *PLoS Genetics* 4 (2): e30.

Guo, Fan, Liying Yan, Hongshan Guo, Lin Li, Boqiang Hu, Yangyu Zhao, Jun Yong, et al., 2015. “The Transcriptome and DNA Methylome Landscapes of Human Primordial Germ Cells.” *Cell* 161 (6): 1437–52.

Lynch, Michael. 2010. “Evolution of the Mutation Rate.” *Trends in Genetics: TIG* 26 (8): 345–52.

Rodríguez-Nuevo, Aida, Ariadna Torres-Sanchez, Juan M. Duran, Cristian De Guirior, Maria Angeles Martínez-Zamora, and Elvan Böke. 2022. “Oocytes Maintain ROS-Free Mitochondrial Metabolism by Suppressing Complex I.” *Nature*, July, 1–6.

Smith, Tegan B., Matthew D. Dun, Nathan D. Smith, Ben J. Curry, Haley S. Connaughton, and Robert J. Aitken. 2013. “The Presence of a Truncated Base Excision Repair Pathway in Human Spermatozoa That Is Mediated by OGG1.” *Journal of Cell Science* 126 (Pt 6): 1488–97.